# Impact of Drying Methods on Phenolic Components and Antioxidant Activity of Sea Buckthorn (*Hippophae rhamnoides* L.) Berries from Different Varieties in China

**DOI:** 10.3390/molecules26237189

**Published:** 2021-11-26

**Authors:** Yue Li, Pei Li, Kailin Yang, Qian He, Yue Wang, Yuhua Sun, Chunnian He, Peigen Xiao

**Affiliations:** 1Institute of Medicinal Plant Development, Chinese Academy of Medicinal Sciences & Peking Union Medicinal College, Beijing 100193, China; 18073049745@163.com (Y.L.); lee_p1214@163.com (P.L.); yangkailin199908@163.com (K.Y.); heqian971003@163.com (Q.H.); wywxc1019@163.com (Y.W.); pgxiao@implad.ac.cn (P.X.); 2Key Laboratory of Bioactive Substances and Resources Utilization of Chinese Herbal Medicine, Ministry of Education, Beijing 100193, China; 3Baotou Medical College, Baotao 014060, China; 4Xinjiang Key Laboratory for Uighur Medicines, Xinjiang Institute of Materia Medica, Urumqi 830004, China; sunyuh1117@aliyun.com

**Keywords:** sea buckthorn, drying method, variety, phenolic compounds, UPLC-Q-TOF-MS, antioxidant activity

## Abstract

Sea buckthorn berries are rich in bioactive compounds and can be used for medicine and food. The variety and drying method used have an important influence on quality. In this study, different sea buckthorn varieties from China were selected and dried with four common drying methods. The total phenolic content (TPC), total flavonoids content (TFC), contents of 12 phenolic compounds and antioxidant capacity in vitro were analyzed. The results showed that the TPC, TFC and antioxidant activity of two wild sea buckthorn berries were higher than those of three cultivated berries, and for the same varieties, measured chemical contents and antioxidant activity of the freeze-dried fruit were significantly higher than those obtained with three conventional drying methods. In addition, forty-one compounds in sea buckthorn berry were identified by UPLC-PDA-Q/TOF-MS, most of which were isorhamnetin derivatives. Multivariate statistical analysis revealed narcissin and isorhamnetin-3-*O*-glucoside varied significantly in sea buckthorn berries of different varieties and with different drying methods; they were potential quality markers. Strong correlations were found between TPC, gallic acid and antioxidant capacity (*p* < 0.05). The results revealed how components and antioxidant activity varied in different sea buckthorn, which provides a valuable reference for quality control and further development and utilization of sea buckthorn.

## 1. Introduction

Sea buckthorn (*Hippophae rhamnoides* L.), belonging to the family Elaeagnaceae, is a plant with important ecological and economic value [1]. With the huge demand of consumers for a healthy natural diet, sea buckthorn berries are attracting increasing attention because of their rich contents of a variety of bioactive compounds [2]. In China, Russia, Mongolia, Turkey and many other places, sea buckthorn is used as a folk medicine to treat stomach ulcers, cough, skin diseases, jaundice, asthma, high blood pressure, rheumatism and genital inflammation [2,3]. In addition to being used as a pharmaceutical, sea buckthorn can be used in food and cosmetics [4]. Modern research has shown that sea buckthorn berries are rich in a variety of bioactive constituents and nutrients: hydrophilic compounds (phenolic acids, flavonoids, ascorbic acid), lipophilic antioxidants (vitamins, carotenoids, tocopherols), protein and amino acids, lipid compounds and fatty acids, sugars and polysaccharides, mineral elements, etc. [5,6]. The berry has been shown to possess antioxidant, antibacterial, cytoprotective, antitumor, anti-inflammatory, immunomodulatory, antifatigue and anticholinergic activities [3,4,6,7,8,9,10,11,12]. The importance of sea buckthorn is often attributed to its high levels of natural antioxidants, of which polyphenols and flavonoids are important components [1,8,13]. Moreover, the widespread application of sea buckthorn in dermatology may be related to the fact that sea buckthorn contains a variety of flavonoids [2]. Flavonols isolated from sea buckthorn have shown great potential in supporting heart and vascular system health [13].

Sea buckthorn is native to Eurasia [2]. It is estimated that the global area of sea buckthorn is approximately 3 million hectares, of which 85 percent are in China [6]. India, Mongolia, Europe, Russia and other places also have a large distribution. *Hippophae rhamnoides* ssp. *sinensis*, endemic to China, is widely distributed in Shanxi, Hebei, Inner Mongolia, Qinghai, Xinjiang, Sichuan and other provinces. As the main source of sea buckthorn, *Hippophae rhamnoides* ssp. *sinensis* exhibits abundant variation in morphological characteristics and secondary metabolites [4]; in addition, there are more than 150 cultivars in the world according to incomplete statistics [14]. However, there has been no systematic comparison of the chemical composition and biological activity of sea buckthorn from different sources in China, which may have a potential impact on the product quality of sea buckthorn [13,15].

Unlike smaller berries such as blueberries, fresh sea buckthorn berries are brittle and difficult to transport. For practical production, in addition to some direct production, fresh sea buckthorn fruit should be dried before application in many cases. In the 2020 edition of the Chinese Pharmacopoeia, sea buckthorn is still processed by traditional drying methods, including hot air drying, sun drying or shade drying (drying in a ventilated place without light). Furthermore, 90% of enterprises use hot air drying given its simple operation and low cost [16]. Sun drying and shade drying do not require equipment or manpower and are still widely used in China. The stability of valuable bioactive substances in sea buckthorn berries will be affected during the dehydration process. Previous studies showed that heat treatment could degrade flavonoids in sea buckthorn extract and reduce the antioxidant activity of the extract [17]. Freeze drying can preserve vitamin C, total phenols and total carotenoids in sea buckthorn berries to a greater extent [18]. Therefore, it is vital to choose the appropriate drying method to prevent possible decomposition of phytochemicals and microbial contamination [19]. However, the extent of changes in the composition and activity of sea buckthorn fruits obtained by different drying methods is unclear, especially the changes in the relative composition of flavonoids and phenolic acids, which also have a significant impact on the quality of sea buckthorn.

In this paper, four commonly used methods were employed to dry sea buckthorn produced in China, including wild sea buckthorn with its abundant resources and cultivated varieties commonly used in the market. The contents of total polyphenols, total flavonoids and 12 components in sea buckthorn samples were determined, and their antioxidant capacity was also detected. The main compounds were identified by UPLC-PAD-Q/TOF-MS. The results showed that different sources and drying methods had significant effects on antioxidant activity of sea buckthorn, and revealed which components were significantly varied in different sea buckthorn. The results can be used for the identification and quality control of sea buckthorn varieties and provide data reference for further utilization, for example, designing some functional products with good antioxidant activity.

## 2. Materials and Methods

### 2.1. Plant Material and Reagents

Five fresh sea buckthorn berry varieties were harvested in October 2020 in China (Figure 1). Two wild *Hippophae rhamnoides* ssp. *sinensis* species were harvested in Lanxian, Shanxi Province (W1) and Zhangbei, Hebei Province (W2). Three cultivated sea buckthorn species were collected, namely, Zhuangyuanhuang (ZYH; E’min, Xinjiang), Shenqiuhong (SQH; Tieling, Liaoning) and Zayou No. 1 (SG; Shanxi, Lanxian). The longitudinal diameter of the berries of the three cultivars was 1.22 to 2.13 times that of the wild varieties (Appendix A). The berry samples were randomly picked from more than three shrubs, mixed for each variety, and stored at −20 °C until processing and analysis. In addition, a commercial dried berry of sea buckthorn, labelled C, was purchased in a market as a reference.

The purity of 14 standards was above 98%: gallic acid **(GA)**, protocatechuic acid **(PA)**, rutin **(RU)**, isoquercitrin **(Q3G)**, isorhamnetin-3-*O*-neoshesperidoside **(I3N)**, quercetin **(QE)**, kaempferol **(KA)**, isorhamnetin **(IS)** and catechin were purchased from Shanghai Yuanye Biological Science and Technology Co., Ltd. (Shanghai, China); kaempferol-3-*O*-rutinoside **(K3R)**, astragalin **(K3G),** narcissin **(I3R)**, isorhamnetin-3-*O*-glucoside **(I3G)** and neochlorogenic acid were obtained from Chengdu Push Biotechnology Co., Ltd. (Chengdu, China). Folin–Ciocalteu’s phenol reagent, 2,2-diphenyl-1-picrylhydrazyl (DPPH) and Trolox were purchased from Sigma–Aldrich (St. Louis, MO, USA). Total antioxidant capacity assay kits (ABTS assay and FRAP assay) were purchased from Beyotime Institute of Biotechnology (Shanghai, China). HPLC-grade methanol and acetonitrile were purchased from Honeywell (Morristown, NJ, USA). Other reagents were analytically pure and purchased from Aladdin (Shanghai, China). Experimental ultrapure water was produced with a Milli-Q system (Millipore Corp, Bedford, MA, USA).

### 2.2. Drying Processes

Four drying methods were used to process sea buckthorn berries: (1) berries were homogenized and lyophilized (−55 °C, 0.20 mbar) for 72 h in a freeze-dryer (Lyovapor^TM^ L-200, BUCHI Labortechnik AG, Flawil, Switzerland) (**L**); (2) berries were dried at 50 °C for five days in a laboratory hot air drying oven (DGG-9070B, Sumsung Laboratory Instrument Co., Ltd., Shanghai, China) (**H**); (3) berries are naturally dried by sunlight for one month at an ambient temperature of 25 °C with an average humidity of 20%. (**S**); (4) berries were kept in a ventilated and dark place to dry naturally, and drying time, environment temperature and average humidity were the same as (3) (**D**). After drying, the berries were crushed evenly and stored in a sealable plastic bag at −20 °C. The samples were weighed accurately before and after drying, and the loss of water was calculated to ensure that water loss with the different drying methods was the same.

### 2.3. Preparation of Samples for Analysis

The extraction method used to determine total phenolic content (TPC), total flavonoids content (TFC) and antioxidant capacity was the same. That is, berry powder (0.40 g) was extracted with 10 mL of methanol: water (5:5, *v*/*v*) in an ultrasonic water bath (KunShan Ultrasonic Instruments Co., Ltd., Kunshan, China) at 50 °C (200 W, 40 kHz) for 40 min. Complementing the weight loss after the mixture was cooled to room temperature, it was filtered through a 0.22 μm nylon microfilter (Jinteng Experimental Equipment Co., Ltd. Tianjin, China).

For determination of individual contents of phenolic components, to quantify more low-concentration compounds, the solid–liquid ratio was 1:10, and other conditions were consistent with the above method.

### 2.4. Determination of Total Phenolic (TPC) and Total Flavonoids (TFC) Contents

TPC was measured by the Folin–Ciocalteu method reported by Orsavová [20]. Briefly, 100 µL of sea buckthorn extract or a series of concentrations of gallic acid solution was added to 4 mL of water, 0.25 mL of F-C reagent and 0.75 mL of 20% Na_2_CO_3_ solution (*w*/*v*); after fully mixing, the mixture was incubated for 60 min in the dark, and absorbance was measured at 750 nm using an Infinite M200 Microplate Reader (TECAN, Männedorf, Switzerland). The standard curve of gallic acid was drawn. The results were expressed as milligrams of gallic acid equivalents per gram of dried sample (mg GE·g^−1^).

TFC was determined using NaNO_2_, Al(NO_3_)_3_ and NaOH according to the Chinese Pharmacopoeia (2020) with slight modification. Two milliliters of berry extract or an appropriate volume of rutin solution was added to 50% (*v*/*v*) methanol aqueous solution, and the mixture volume was 5.2 mL. After mixing, 0.4 mL of 5% NaNO_2_ solution was added and reacted for 6 min, then 0.4 mL of 10% Al(NO_3_)_3_ solution was added and reacted for 6 min, 4 mL of 10% NaOH solution was added and reacted for 15 min after shaking, and the solution was detected at a wavelength of 500 nm. The standard curve of rutin was drawn. The results were expressed as milligrams of rutin equivalents per gram of dried sample (mg RE·g^−1^).

### 2.5. Quantitative Determination of 12 Compound Contents by UPLC-DAD

Quantification of 12 phenolic compounds was performed by a Thermo Ultimate 3000 UHPLC system equipped with a DAD-3000RS (Thermo Fisher Scientific, Germering, Germany). A Waters ACQUITY UPLC BEH C18 (2.1 × 100 mm, 1.7 µm) column was used. The mobile phase consisted of formic acid-water (0.1:100, *v*/*v*) (A) and acetonitrile (B), and the gradient elution procedure was as follows: 0–3 min, 7–9% B; 3–8 min, 9–13% B; 8–11 min, 13% B; 11 to 18 min, 13–17% B; 18–23 min, 17% B; 23–30 min, 17–35% B; 30 –31 min, 35–100% B; 31–31.5 min, 100% B; 31.5–32 min, 100–7% B; 32–34 min, 7% B. The flow rate was 0.32 mL/min, and the injection volume was 5 µL. The temperature of the column and the sample tray was 30 °C and 10 °C, respectively. The detection wavelengths were set at 360 nm and 254 nm. Chromeleon 7 software was used to acquire and analyze the data.

A mixture stock standard solution containing 0.112 mg/mL **(GA)**, 0.050 mg/mL **(PA)**, 0.632 mg/mL **(RU)**, 0.172 mg/mL **(Q3G)**, 0.114 mg/mL **(I3N)**, 0.121 mg/mL **(K3R)**, 0.125 mg/mL **(K3G)**, 0.674 mg/mL **(I3R)**, 0.331 mg/mL **(I3G)**, 0.031 mg/mL **(QE)**, 0.023 mg/mL **(KA)** and 0.029 mg/mL **(IS)** was diluted by 2, 4, 8, 16, 32, 64 and 128 times, to obtain standard solutions that could be used to plot standard curves. Prior to UPLC analysis, all solutions were stored at 4 °C and filtered through 0.22 μm nylon micropore membranes.

### 2.6. Identification of Phenolic Compounds by UPLC-PDA-Q/TOF-MS

The analysis was carried out with a Waters Acquity ultrahigh-performance LC system (Waters Corp., Milford, MA, USA) with a PDA detector coupled to a Xevo G2 Q/TOF micromass spectrometer (Waters Corp., Milford, MA, USA) equipped with an electrospray ionization (ESI) source. The extraction protocols and UPLC parameters were the same as in the UPLC-DAD analysis. The optimized MS parameters were as follows: negative ionization mode; mass scanning range of 50–1200 Da; cone voltage of 40 V; capillary voltage of 2.5 kV; ion source temperature of 100 °C; desolvation temperature of 250 °C; desolvation gas flow rate of 600 L/h; and cone gas flow rate of 50 L/h. MassLynx (version 4.1, Waters Corp., Milford, MA, USA) was used to collect and process data.

Progenesis QI 2.0 software (Waters Corp., Milford, MA, USA) was used to identify compounds on-line, and its workflow included peak alignment, peak picking, normalization and deconvolution. To improve the reliability of compound identification, compounds in sea buckthorn berry were collected from literature, and an in-house compound database of 121 compounds was established. The identification results were obtained by comparing online identification with local database information characteristics.

### 2.7. Antioxidant Activity by DPPH, ABTS and FRAP Assays

The DPPH radical scavenging assay was established according to a previous report [21] and was slightly modified. Briefly, 40 μL of diluted sample solution was added to 160 μL of freshly prepared 0.2 mM DPPH solution and reacted for 30 min at 37 °C in the dark, and absorbance was measured at 517 nm. The standard curve of Trolox was drawn. The antioxidant capacity of the samples was expressed as milligrams of Trolox equivalents per gram of dried sample (mg TE·g^−1^).

The total antioxidant capacities were evaluated using the ABTS method and FRAP method described by Zhang [22] and Luo [23]. Absorbance was measured at 734 nm and 593 nm. The results were expressed as mg TE·g^−1^.

The overall antioxidant activity of the sea buckthorn sample was evaluated by the antioxidant potency composite (APC) index, and the calculation formula was shown in the previous literature [24].

### 2.8. Data Analysis

All experiments were repeated three times, and the results are expressed as the mean ± standard deviation. One-way analysis of variance (ANOVA, *p* < 0.05) with Tukey’s HSD and the Tamhane tests was carried out using SPSS 23.0 (SPSS Inc., Chicago, IL, USA). Pearson correlation coefficients were calculated using SPSS 23.0. Principal component analysis (PCA) and orthogonal partial least squares discriminant analysis (OPLS-DA) were carried out using SIMCA Software 14.1 (Umetrics AB, Umea, Sweden).

## 3. Results and Discussion

### 3.1. Total Phenolic Content (TPC) and Total Flavonoids Content (TFC)

Polyphenols play an essential role in the antioxidant activity of sea buckthorn, and their contents varied greatly among different varieties [25]. In the investigated specimens, W1 and W2 presented obviously higher amounts of TPC and TFC (Table 1). The TPC of lyophilized berries of W1 and W2 were 33.51 and 32.20 mg GAE·g^−1^, respectively, which were higher than the published values of 2.05 and 2.45 mg GAE·g^−1^ in lyophilized berries in Terhi and Tytti from Finland [8], and also higher than the TPC of nine cultivars of sea buckthorn from Slovakia (5.19–23.97 mg GAE·g^−1^ DW) [17]. According to Ficzek, the TPC of Leikora and Askola (the German sea buckthorn cultivar) and Orangeveja (the Siberian cultivar) were 1.86, 2.95 and 3.81 mg GAE·g^−1^, respectively [25]. Such variability in TPC may be due to different extraction procedures, genetic backgrounds and locality types. High amounts of TFC were measured in lyophilized sea buckthorn with values of 1.98–8.96 mg RE·g^−1^, which was significantly higher than the TFC of nine sea buckthorn specimens from Slovakia (4.37–27.54 mg RE·kg^−1^) [1]. In addition, The TPC and TFC of W1 and W2 are higher than those of C (8.48 mg GAE·g^−1^ and 1.81 mg RE·g^−1^, respectively), a popular commercial variety. The results suggest wild *Hippophae rhamnoides* ssp. *sinensis* may be a good germplasm resource due to its high content of total polyphenols and flavonoids, and is worth further study and utilization.

Among the different drying methods, the TPC of freeze-dried berries, ranging from 11.80 mg GAE·g^−1^ to 33.51 mg GAE·g^−1^, was 1.56–2.97 times that of other dried berries. Because of prolonged processes of hot air drying, sun drying and shade drying, phenolic compounds are easily degraded by polyphenol oxidase. Moreover, the high temperature of hot air drying may also destroy polyphenolic components [18]. Regarding the effect of the other three drying methods on TPC, significant differences were observed in W1, W2, SQH and SG (*p* < 0.05). The TPC of hot air-dried berries was higher than that of sun-dried and shade-dried berries in ZYH, SQH and SG, which may indicate that the effect of high temperature is lower than drying time in terms of destroying polyphenols in sea buckthorn. The same situation was observed for total flavonoids content, and freeze-dried berries still maintained their dominant position, so freeze drying can retain the active ingredients in sea buckthorn berries to the maximum extent [18,19,26]. However, the difference was not as significant as that for TPC; especially for the SG cultivar, and the four drying methods had no significant effect on their TFC, which indicates that different drying methods had lower effects on TFC than TPC, and total flavonoids content might be relatively stable.

### 3.2. Quantitative Determination of Twelve Components in Sea Buckthorn Berries

An analytical UPLC-DAD method for simultaneous determination of the 12 bioactive components in sea buckthorn berries was developed. It was validated in terms of defining the linearity, limits of quantification (LOQ) and detection (LOD), precision, repeatability, stability and recovery; the results showed that the method was good (Appendix A). The method has been successfully applied in sea buckthorn berries. UPLC-DAD chromatograms of representative sample W1 and mixed standard solution are shown in Appendix A. The component contents of 21 sea buckthorn samples are shown in Appendix A. The results showed that there were significant differences in the content of the same compound in different samples, which was probably due to different varieties, processing methods and growing locations.

In general, the total contents of 12 compounds were notably highest in SG, followed by ZYH, W2, W1 and SQH, which was probably due to different genetic backgrounds. For the same variety, the total content of 12 compounds in lyophilized berries was significantly higher than that of the other three (*p* < 0.05), which may be because lyophilization provides a low-temperature, no-oxygen environment to retain the active components in sea buckthorn to the greatest extent, making it the closest to fresh fruit [27]. Regarding the other three drying methods, in wild W1 and W2, the total contents of hot air-dried berries were significantly lower than those of sun-dried and shade-dried berries. In contrast, in ZYH and SG, the total contents of hot air-dried berries were highest. The results showed that the effects of different drying methods on the content might have a certain correlation with varieties.

According to previous reports, isorhamnetin glycosides and quercetin glycosides are the most important flavonols in sea buckthorn berries [6,28]. In this study, the highest content was narcissin in all samples, followed by isorhamnetin-3-*O*-glucoside, rutin and isoquercitrin, which is consistent with previous literature reports [5]. In lyophilized berries of SG, the content of narcissin (377.25 mg·100 g^−1^) is particularly high, which is 2.1–3.7 times that of the other four kinds of sea buckthorn (101.57–178 mg·100 g^−1^). Tkacz [6] reported that the content of narcissin in four sea buckthorn cultivars from Poland ranged from 96.4 to 228 mg·100 g^−1^. Ma [15] reported that the contents of narcissin in *H. rhamnoides* ssp. *mongolica* and *H. rhamnoides* ssp. *sinensis* were 15 and 38 mg·100 g^−1^, respectively. The high content of narcissin may become an important indicator to distinguish SG from other sea buckthorn varieties. Isorhamnetin-3-*O*-glucoside had high amounts ranging from 23.07 (W2) to 83.04 mg·100 g^−1^ (SG). Yang [13] reported the content range of isorhamnetin-3-*O*-glucoside in Finnish cultivars (5.9–25.9 mg·100 g^−1^), Finnish wild varieties (5.4–25.8 mg·100 g^−1^), Russian cultivars (1.8–12.5 mg·100 g^−1^) and Chinese wild sea buckthorn (4.8–18.9 mg·100 g^−1^), they are significantly lower than five sea buckthorn that we tested.

In addition to flavonoids, phenolic acids in sea buckthorn are noteworthy. Gallic acid has been reported to have a wide range of biological activities [29,30], and it was recorded with the lowest content of 27.41 mg·100 g^−1^ in cultivar SG and the highest content of 61.32 mg·100 g^−1^ in W2; they were higher than the content of gallic acid in four Romanian sea buckthorn (6.51–19.37 mg·kg^−1^) [31]. The content range of protocatechuic acid was 0.1–3.5 mg·kg^−1^ in sea buckthorn from Slovakia [1]. In lyophilized berries, protocatechuic acid was only detected in W2, with a value of 1.42 mg·kg^−1^. Notably, the protocatechuic content in freeze-dried berries was significantly lower than that in the berries treated by hot air drying, sun drying and shade drying. It may be because the other three drying methods took longer than lyophilization, during which some components (possibly protocatechuic acid ethyl ester) in sea buckthorn were converted to protocatechuic acid [1].

As shown in Appendix A, although several high peaks were quantified in sea buckthorn samples, there were still other high peaks that were not identified; in other words, 12 components do not represent the overall chemical characteristics of sea buckthorn. Therefore, more compounds in sea buckthorn were identified by UPLC-PDA-Q/TOF-MS (Section 3.3), and marker compounds of different varieties and different methods were recognized using multivariate statistical analysis (Section 3.4).

### 3.3. Identification of Phytochemicals

Sea buckthorn samples W1 and SQH were selected and analyzed by UPLC-PDA-Q/TOF-MS. The identification methods were as follows: (1) retention times, UV spectra, and mass spectrometry fragments were compared with those of authentic standards; (2) an in-house database of sea buckthorn was established, and accurate molecular mass and MS/MS fragment ions were compared with the established database and open-source databases in Progenesis QI. The error between the measured mass and theoretical mass was set to <12 ppm. Finally, 41 compounds were tentatively identified, most of which were flavonol derivatives. The results are shown in Figure 2 and Table 2.

Four phenolic acids were identified as gallic acid (**1**), protocatechuic acid (**2**), neochlorogenic acid (**3**) and hydroxycinnamic acid (**6**), and the first three (**1**, **2** and **3**) were compared with reference standards. The UV absorption bands of compound **6** (Rt = 4.13 min) were 230 nm and 276 nm, which showed long conjugated groups; furthermore, the molecular ion peak at *m*/*z* 163.0390 and the fragment ion at *m*/*z* 120.05 [M-COO^−^]^−^ indicated that **6** corresponded to hydroxycinnamic acid, but the position of the hydroxy group on the benzene ring was uncertain. In addition, three catechin derivatives, including **4** (epigallocatechin) and **5** (catechin), were confirmed based on comparison with standard substances, and **9** was provisionally identified as epicatechin based on precise molecular mass and relative retention time.

Thirty-four flavonol derivatives, including six quercetin derivatives (**7, 10, 16, 18, 21** and **22**), eight kaempferol derivatives (**11, 12, 20, 25, 26, 27, 31** and **33**), fourteen isorhamnetin derivatives (**13, 14, 15, 17, 19, 23, 24, 28, 29, 30, 32, 34, 35** and **36**), two syringetin derivatives (**37** and **38**) and four aglycons (**8, 39, 40** and **41**), were preliminarily identified. Among them, **22, 23, 29, 31, 33, 34, 36, 39, 40** and **41** were confirmed based on comparison with standard substances. The characteristic wavelengths of flavonols at band I (328–385 nm) and band II (250–280 nm) were observed. The mass-to-charge ratios of quercetin, kaempferol, isorhamnetin and syringetin were 301.03, 285.04, 315.05, and 345.06, respectively. The common sugar residues include glucose, rhamnose, sophorose, rutinose, etc., and positions 3 and 7 of aglycones are preferential glycosylation positions [6]. Referring to previous reports, the positions of substituents of some compounds have been preliminarily deduced (**8**, **10**, **11**, **25**, etc.), but there are some compounds whose substituent positions still have not been determined (**7, 19, 26, 35**).

According to previous research on sea buckthorn extracts, glucose (*m*/*z* 162) was the main hexose on flavonol glycosides [2,5,15,34]. Compound **7** (Rt = 4.34 min and [M−H]^−^ at *m*/*z* 625.1393) quercetin-*O*-dihexoside can be conclusively identified quercetin-*O*-diglucoside [32]. The situation is the same for **11, 20, 24, 32, 35,** and **38**. Compound **38** (Rt = 17.05 min) with a molecular ion at *m*/*z* 507.1142 and a fragment ion at *m*/*z* 344.05 was considered to be syringetin-3-*O*-hexoside, which had not been previously reported in sea buckthorn [37,42]. Moreover, the relative retention time can also be used for putative identification. Compounds **16** and **18** had the same MS/MS fragmentation [M-H-146-162]^−^; according to previous literature, in LC analyses using a C18 column, quercetin-3-*O*-glucoside-7-*O*-rhamnoside eluted before quercetin-3-*O*-rhamnosyl-glucoside and quercetin-3-*O*-rutinoside [6,28]. Moreover, **18** did not have a fragmentation ion at *m*/*z* 477.10, indicating that it may lose both rhamnose and glucose directly at the same time. Thus, these compounds were preliminarily concluded to be quercetin-3-*O*-glucoside-7-*O*-rhamnoside [6,32] and quercetin-3-*O*-rhamnosyl-glucoside [26], respectively. The loss of 146 Da may also indicate -courmaroyl, and the loss of 206 Da indicated -sinapoyl. Compound **27** (Rt = 12.49 min and [M−H]^−^ at *m*/*z* 961.2596) was tentatively identified as kaempferol-3-*O*-(6-*O*-sinapoyl)glucose-glucoside-7-*O*-rhamnoside, which had ions at *m*/*z* 815.20 [M-H-rhamnose]^−^, *m*/*z* 609.14 [M-H-rhamnose-sinapoyl]^−^ and *m*/*z* 431.10 [M-H-sinapoyl-diglucose]^−^ [41].

### 3.4. Marker Compounds in Berries of Different Varieties and Different Drying Methods

To comprehensively analyze the differences in specific components in different varieties of sea buckthorn and with different drying methods, UPLC-UV combined with multivariate statistical analysis [43] was performed. Sixty chromatographic datasets were analyzed by Chem Pattern (Chemmind Technologies Co., Ltd., Beijing, China). A lyophilized sample of W1 (W1-L1) was designated as the representative sample, and the peaks with retention times between 1.3 and 30 min and relative peak areas >0.1% were screened out. Finally, 73 peaks were screened out, as shown in Appendix A. Peak alignment was performed with the peaks of the reference standards.

In our study, first, the UPLC-DAD chromatographic matrix of compounds (60 samples* 73 peak variables) was subjected to HCA analysis using SIMCA.14.1 software (Figure 3). Firstly, wild sea buckthorn W1 and W2 clustered into one class, while ZYH, SQH and SG clustered into another class; it is evident that the genetic background may have a more critical effect on the composition and content of phenolic compounds than the drying method. Among the three cultivated sea buckthorns, SG is a hybrid of *Hippophae rhamnoides* ssp. *sinensis* and *Hippophae rhamnoides* ssp. *mongolica*, and ZYH and SQH were bred from the same batch of sea buckthorn seeds by Huang [44], so the latter two aggregate more closely. Among the different drying methods, lyophilization was clearly distinguished from the other three drying methods in all samples. For samples W1, W2, SG and SQH, sun-dried and shade-dried berries were clustered into a small category, which was distinguished from the hot air-dried berries. The possible reasons are as follows: the processes of sun drying and shade drying were slow, and metabolism might last longer, leading to the degradation of some key compounds and then to the loss of bioactive components in the extract [45]; additionally, the high temperatures used to dry sea buckthorn berries may also destroy their chemical composition.

To compare the different compounds between different drying methods, wild sea buckthorn W1 with high contents and good antioxidant activity was taken as an example, and its UPLC-DAD data (12 samples* 73 peaks) was used for principal component analysis (PCA). The sum of PC1 and PC2 is 85.4%, indicating that the model is reliable and stable. As shown in Figure 4A, sun-dried and shade-dried samples were grouped into one group (**S&D** group), while lyophilized (**L**) and hot air-dried (**H**) samples were grouped into separate groups. Therefore, pairwise OPLS-DA was performed on the **L**, **H** and **S****&D** groups to identify the different compounds between different drying groups. S-plots is shown in Figure 4B–D; peaks whose VIP > 2 and *p* < 0.05 were screened out as high-contributed variables, so the different compounds between the **L** and **H** groups were peaks 46, 12, 21 and 47, which were preliminarily identified as narcissin, kaempferol-3-*O*-sophoroside-7-*O*-rhamnoside, isorhamnetin-3-*O*-rutinoside-7-*O*-rhamnoside and isorhamnetin-3-*O*-glucoside. In the same way, the different compounds between the **L** and **S&D** groups were peaks 46 (narcissin) and 47 (isorhamnetin-3-*O*-glucoside). The different compounds between the **H** and **S&D** groups were peaks 12 (kaempferol 3-*O*-sophoroside-7-*O*-rhamnoside), 46 (narcissin), 21 (isorhamnetin 3-*O*-rutinoside-7-*O*-rhamnoside) and 16 (isorhamnetin-3-*O*-sophroside-7-*O*-rhamnoside).

For the comparison of different compounds among different sea buckthorn varieties, wild sea buckthorn W1 and W2 were divided into one group, and ZYH, SQH and SG were divided into the other group. The OPLS-DA score showed that wild berries and cultivated berries could be completely distinguished, as shown in Figure 5. The differential compounds in the S-plot are peaks 46 (narcissin), 47 (isorhamnetin-3-*O*-glucoside) and 33 (unknown).

The common differential compound for pairwise comparison is narcissin and isorhamnetin-3-*O*-glucoside, indicating that these two compounds were easily influenced by species and drying method. In our previous study (Section 3.2) and reported in the literature [5,13], the levels of these two compounds were exceptionally high and varied considerably from sample to sample. These results suggest that they may be used to control the quality of sea buckthorn. In addition, the aglycone of these two compounds (isorhamnetin) has been reported to have many pharmacological effects, such as antitumor, anti-oxidation, organ protection [46,47]. Moreover, isorhamnetin-3-*O*-glucoside has been reported to have the activity of inhibiting α-glucosidase and enhancing insulin secretion [48]. Therefore, narcissin and isorhamnetin-3-*O*-glucoside are potential quality markers of sea buckthorn [49].

### 3.5. Correlations between Antioxidant Activity (AOA) and Phenolic Compounds

Antioxidant activity was evaluated by DPPH, ABTS and FRAP assays, and their results are displayed in Table 3. The APC index is shown in Appendix A. W1 and W2 performed better antioxidant activity than other cultivars, and lyophilization was the best way to retain antioxidant activity, which was consistent with the results of TPC and TFC. The DPPH values of the five varieties were 13.12–86.30 mg TE·g^−1^, which were higher than those of the other three drying methods (*p* < 0.05). The DPPH values of W1 and W2 were 71.80 mg TE·g^−1^ and 86.30 mg TE·g^−1^, respectively, which were all higher than those of nine sea buckthorn varieties from Slovakia (7.94–29.56 mg TE·g^−1^) [1], and the highest value was approximately 10 times higher. According to Ye’s report, sea buckthorn leaf and berry performed good antioxidant activity, but the leaf was better [8]. The DPPH values of shade-dried berries ranged from 3.64 mg TE·g^−1^ to 29.27 mg TE·g^−1^, significantly higher than those obtained by hot air drying and sun drying. It indicated that high temperature might reduce the antioxidant activity of sea buckthorn, which was consistent with Ma’s study [45]. In W1, SG and SQH, hot air-dried fruit was significantly higher than that in sun-dried fruit (*p* < 0.05), which may be due to loss of active ingredients caused by the longer drying process. For the ABTS and FRAP results, the trend of antioxidant values was basically the same as that of DPPH, indicating that these three methods confirm each other.

To explore the relationship between antioxidant activity and TPC, TFC and individual phenolic compounds, Pearson’s correlation coefficients (r) were calculated, and the results are shown in Table 4. The strong antioxidant capacity of sea buckthorn samples may be explained by the strong correlation between TPC and antioxidant value (r = 0.980, 0.986 and 0.988, respectively). Pearson’s correlation coefficients between the total flavonoids content and antioxidant values (r = 0.577, 0.664, 0.727, respectively) were lower than those for TPC, which is consistent with previous reports [1,50]. For individual compounds, the strongest correlation with antioxidant activities was gallic acid (r = 0.740, 0.713 and 0.734, respectively) [30]. Isorhamnetin-3-*O*-neohesperidoside (r = 0.444, 0.546 and 0.547) and kaempferol-3-*O*-rutinoside (r = 0.397, 0.497, 0.541) appear to have moderate correlations. In addition, SG with high contents of gallic acid had the lowest antioxidant activity, which indicates that the antioxidant activity of sea buckthorn is a combination of multiple compounds. In other words, extracts with the highest abundance of a particular compound did not have stronger antioxidant effects, but rather extracts with a balanced content of bioactive compounds had the highest antioxidant effects [31].

## 4. Conclusions

In this study, the main chemical composition and antioxidant activity in vitro of sea buckthorn, including different varieties and berries dried with different methods, were detected. Two wild sea buckthorns had higher TPC, TFC and antioxidant capacity than three cultivated sea buckthorns. Compared with hot air drying, sun drying and shade drying as recorded in the Chinese Pharmacopoeia (2020), lyophilization is a better method to retain active components and antioxidant activity. Total polyphenols are more likely to be antioxidants than total flavonoids. Among the 12 quantitative components, narcissin was the highest in all samples, particularly in SG, and gallic acid was a potential antioxidant. Meanwhile, a total of 41 compounds were identified by UPLC-PDA-Q/TOF-MS, and the main compounds were isorhamnetin derivatives. Furthermore, UPLC-UV combined with multivariate statistical analysis was used to recognize marker compounds. The different compounds between wild and cultivated sea buckthorn were narcissin and isorhamnetin-3-*O*-glucoside, and the different compounds among different drying methods were narcissin, isorhamnetin-3-*O*-glucoside, kaempferol-3-*O*-sophoroside-7-*O*-rhamnoside, isorhamnetin-3-*O*-rutinoside-7-*O*-rhamnoside and isorhamnetin-3-*O*-sophroside-7-*O*-rhamnoside. These differential compounds may become Q-markers for quality control of sea buckthorn.

Our study provided valuable data for the selection of processing methods and economical crop varieties of sea buckthorn, as well as the identification and quality control of sea buckthorn. It benefits for design of functional products rich in phenolic components with antioxidant activity.

## Figures and Tables

**Figure 1 molecules-26-07189-f001:**
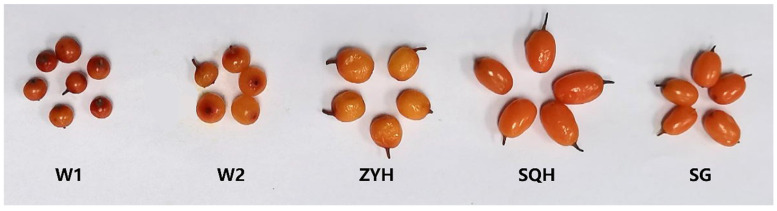
Sea buckthorn berries examined in present study.

**Figure 2 molecules-26-07189-f002:**
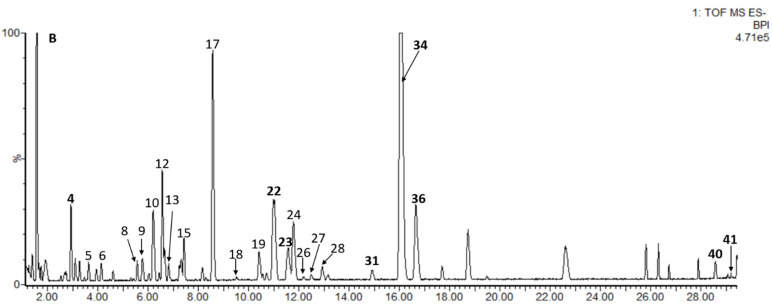
Base peak intensity (BPI) chromatogram of W1 sample in negative ion mode, and the identification of peak numbered **1**–**41** is given in Table 2.

**Figure 3 molecules-26-07189-f003:**
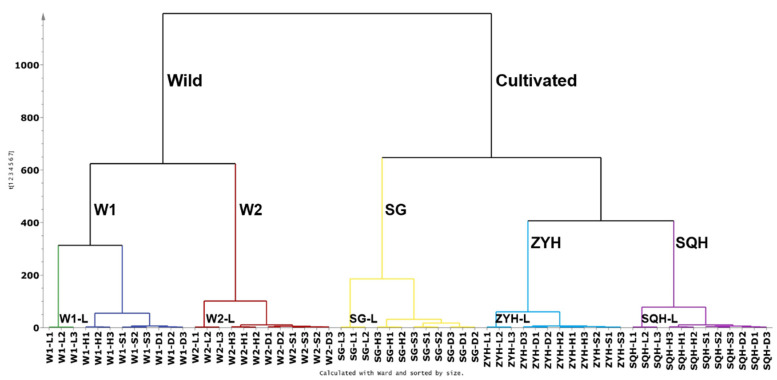
Dendrogram showing the hierarchical clustering results for 21 batches of Sea buckthorn berries.

**Figure 4 molecules-26-07189-f004:**
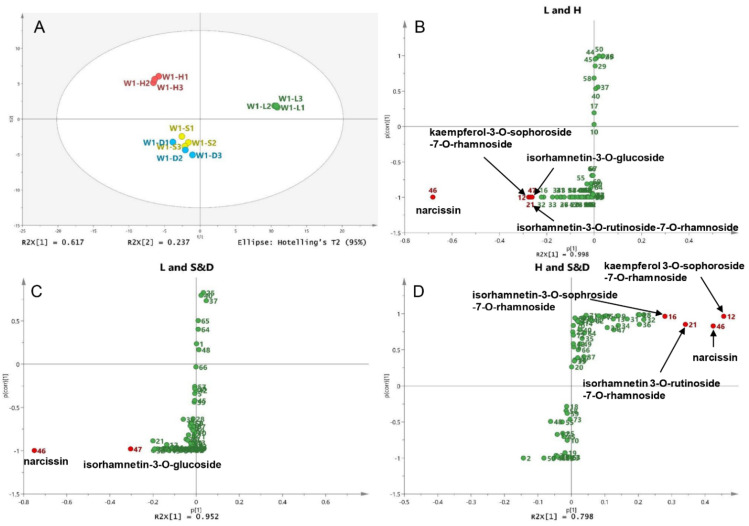
Different compounds in berries of different drying methods: (**A**) Score plot of the PCA for the W1 sample; (**B**) S-plot of the OPLS-DA between the **L** and **H** groups, (**C**) between the **L** and **S**&**D** groups, (**D**) between the **H** and **S**&**D** groups.

**Figure 5 molecules-26-07189-f005:**
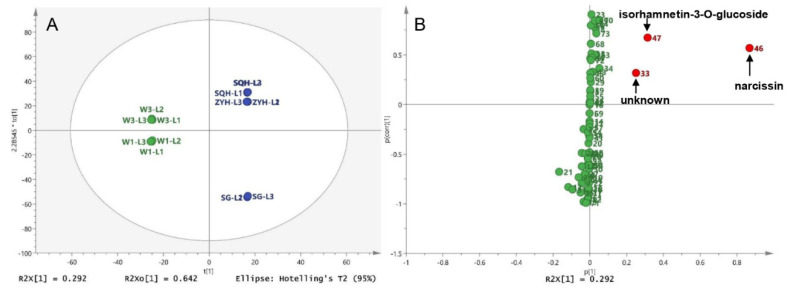
Different compounds between wild and cultivated sea buckthorn: (**A**) Score plot of the OPLS-DA for five lyophilized berries; (**B**) S-plot of the OPLS-DA between wild berries and cultivated berries.

**Table 1 molecules-26-07189-t001:** Contents of total polyphenols (TPC) and total flavonoids (TFC) in sea buckthorn berries of different varieties and drying methods.

Varieties	TPC (mg GAE·g^−1^)				TFC (mg RE·g^−1^)			
	L	H	S	D	L	H	S	D
W1	33.51 ± 0.51 ^a^	14.23 ± 0.13 ^b^	11.28 ± 0.23 ^c^	14.67 ± 0.11 ^b^	8.96 ± 0.45 ^a^	5.36 ± 0.08 ^c^	5.84 ± 0.08 ^c^	6.65 ± 0.23 ^b^
W2	32.20 ± 0.36 ^a^	13.12 ± 0.22 ^c^	15.57 ± 0.07 ^b^	15.26 ± 0.25 ^b^	3.52 ± 0.19 ^b^	4.36 ± 0.13 ^a^	3.50 ± 0.04 ^b^	3.54 ± 0.08 ^b^
ZYH	15.95 ± 0.59 ^a^	7.49 ± 0.23 ^b^	7.18 ± 0.12 ^b^	7.34 ± 0.24 ^b^	2.86 ± 0.05 ^a^	1.72 ± 0.01 ^d^	2.45 ± 0.01 ^b^	2.08 ± 0.02 ^c^
SQH	11.80 ± 0.29 ^a^	7.53 ± 0.04 ^b^	5.32 ± 0.13 ^c^	5.40 ± 0.11 ^c^	2.59 ± 0.08 ^a^	2.35 ± 0.12 ^b^	1.91 ± 0.06 ^c^	1.72 ± 0.03 ^c^
SG	11.83 ± 0.28 ^a^	7.57 ± 0.19 ^b^	7.21 ± 0.11 ^bc^	7.02 ± 0.12 ^c^	1.98 ± 0.09 ^a^	1.95 ± 0.04 ^a^	2.02 ± 0.03 ^a^	2.09 ± 0.03 ^a^
C	8.48 ± 0.22				1.81 ± 0.03			

**L**, **H**, **S** and **D** represent different drying methods: lyophilization, hot air drying, drying under the natural sun and drying in the dark, respectively. Values are expressed as the mean ± standard deviation, n = 3. Significant differences between different drying methods of each variety (*p* < 0.05) are marked a–d. For the same letter, there is no significant difference, and for the opposite, there is a significant difference (*p* < 0.05).

**Table 2 molecules-26-07189-t002:** Detailed information on the putatively identified compounds from sea buckthorn.

No. ^a^	No. ^b^	Rt (min)	λmax (nm)	*m/z*	Adducts	Formula	MS/MS Fragment Ions (*m/z*)	Tentative Identification ^cd^	References
**1**	-	1.23	203, 275	169.0135	M-H	C_7_H_6_O_5_	125.02	Gallic acid *	-
**2**	-	2.11	208, 270	153.0180	M-H	C_7_H_6_O_4_	152.01, 137.02, 121.02	Protocatechuic acid *	-
**3**	-	2.18	208, 276	335.0755	M-H_2_O-H	C_16_H_18_O_9_	305.06, 201.03, 191.05, 125.02	Neochlorogenic acid *	-
**4**	-	2.93	206, 270	305.0643	M-H	C_15_H_14_O_7_	179.03, 125.02	Epigallocatechin *	-
**5**	-	3.63	209, 275	289.0709	M-H	C_15_H_14_O_6_	245.08, 205.05, 125.02	Catechin *	-
**6**	4	4.13	230, 276	163.0390	M-H	C_9_H_8_O_3_	120.05, 119.04, 117.03	Hydroxycinnamic acid	-
**7**	5	4.34	265, 365	625.1393	M-H	C_27_H_30_O_17_	463.08, 301.03	Quercetin-*O*-dihexoside	[32]
**8**	9	5.56	253, 354	771.1980	M-H	C_33_H_40_O_21_	625.14, 609.14, 446.08, 305.06	Quercetin-3-*O*-sophoroside-7-*O*-rhamnoside	[6,28]
**9**	10	5.78	233, 276	289.0710	M-H	C_15_H_14_O_6_	245.08, 203.07, 125.02, 109.02	Epicatechin	[32]
**10**	11	6.21	266, 348	755.2031	M-H	C_33_H_40_O_20_	609.1470	Quercetin-3-*O*-rhamnosyl-glucoside-7-*O*-rhamnoside	[28,33]
**11**	-	6.54	265, 348	639.1567	M-H	C_28_H_32_O_17_	609.14, 477.10, 315.05	Isorhamnetin-3,7-*O*-dihexoside	[6,28]
**12**	12	6.55	265, 348	755.2054	M-H	C_33_H_40_O_20_	609.14, 430.09, 257.04	Kaempferol-3-*O*-sophoroside-7-*O*-rhamnoside	[34,35]
**13**	13	6.81	252, 349	785.2142	M-H	C_34_H_42_O_21_	623.16, 477.10, 315.05	Isorhamnetin-3-*O*-rutinoside-7-*O*-glucoside	[28]
**14**	-	6.82	252, 349	623.1612	M-H	C_28_H_32_O_16_	477.10, 357.06, 315.05	Isorhamnetin-3-*O*-glucoside-7-*O*-rhamnoside	[15]
**15**	16	7.43	253, 353	785.2145	M-H	C_34_H_42_O_21_	639.15, 315.05	Isorhamnetin-3-*O*-sophroside-7-*O*-rhamnoside	[6,28,36]
**16**	-	8.53	255, 353	609.1450	M-H	C_27_H_30_O_16_	463.08, 300.02, 299.01	Quercetin-3-*O*-hexoside-7-*O*-rhamnoside	[6,37]
**17**	21	8.58	254, 348	753.1872	M-H_2_O-H	C_33_H_40_O_21_	591.13, 489.10	Isorhamnetin 3-*O*-rutinoside-7-*O*-rhamnoside	[38]
**18**	25	9.52	260, 348	609.1446	M-H	C_27_H_30_O_16_	300.02, 271.02,	Quercetin-3-*O*-rhamnosyl-glucoside	[28]
**19**	28	10.42	265,348	739.2073	M-H	C_33_H_40_O_19_	576.15, 284.03	Kaempferol-glucoside-dirhamnoside	[37,39]
**20**	-	10.71	263, 348	593.1499	M-H	C_27_H_30_O_15_	447.09, 285.04	Kaempferol-3-*O*-hexoside-7-*O*-rhamnoside	[6]
**21**	-	10.75	246, 342	917.2352	M-H	C_42_H_46_O_23_	771.18, 623.16, 201.04	Quercetin-3-coumaroyl-diglucoside-7-*O*-rhamnoside	[34]
**22**	32	10.98	254, 354	609.1456	M-H	C_27_H_30_O_16_	300.03, 271.02	Rutin *	-
**23**	34	11.58	254, 348	463.0878	M-H	C_21_H_20_O_12_	300.03, 271.02, 255.03	Isoquercitrin *	-
**24**	35	11.78	254, 349	623.1616	M-H	C_28_H_32_O_16_	477.10, 461.11, 315.05	Isorhamnetin-3-*O*-(2-rhamnosyl)hexoside	[6]
**25**	-	11.82	251, 349	477.1022	M+FA-H	C_21_H_20_O_10_	461.11, 285.04	Kaempferol-7-*O*-rhamnoside	[40]
**26**	36	12.18	250, 349	639.1544	M+FA-H	C_27_H_30_O_15_	330.04	Kaempferol-3-glucoside-rhamnoside	[28]
**27**	-	12.49	246, 339	961.2596	M-H	C_44_H_50_O_24_	837.19, 815.20, 431.10, 284.03	Kaempferol-3-*O*-(6-*O*-sinapoyl)glucose-glucoside-7-*O*-rhamnoside	[41]
**28**	37	12.94	246, 341	991.2716	M-H	C_45_H_52_O_25_	845.21, 639.16, 460.10	Isorhamnetin-3-*O*-(6-*O*-sinapoyl)glucose-glucoside -7-*O*-rhamnoside	[28,41]
**29**	38	13.06	253, 349	623.1604	M-H	C_28_H_32_O_16_	314.04	Isorhamnetin-3-*O*-neohesperidoside *	-
**30**	41	13.61	247, 336	931.2506	M-H	C_43_H_48_O_23_	785.20, 639.16, 460.10, 314.04	Isorhamnetin-3-coumaroyl-diglucoside-7-rhamnoside	[34]
**31**	43	14.93	265, 348	593.1490	M-H	C_27_H_30_O_15_	285.04, 255.03	Kaempferol-3-*O*-rutinoside *	-
**32**	44	15.22	252, 347	623.1604	M-H	C_28_H_32_O_16_	314.04, 299.02	Isorhamnetin-3-*O*-(6-rhamnosyl)hexoside	[6]
**33**	45	15.42	265, 348	447.0920	M-H	C_21_H_20_O_11_	284.03, 255.03	Kaempferol 3-*O*-glucoside *	-
**34**	46	16.06	254, 354	623.1628	M-H	C_28_H_32_O_16_	357.06, 315.05, 314.04	Narcissin *	-
**35**	-	16.07	254, 354	639.1557	M-H	C_28_H_32_O_17_	315.05, 314.04	Isorhamnetin-*O*-dihexoside	[6,28]
**36**	47	16.67	253, 349	477.1049	M-H	C_22_H_22_O_12_	314.04	Isorhamnetin-3-*O*-glucoside *	-
**37**	-	16.72	251, 348	653.1713	M-H	C_29_H_34_O_17_	447.09, 345.06	Syringetin 3-*O*-rutinoside	[37]
**38**	48	17.05	250, 348	507.1142	M-H	C_23_H_24_O_13_	344.0	Syringetin-3-*O*-hexoside	[37,42]
**39**	61	24.82	254, 363	301.0369	M-H	C_15_H_10_O_7_	-	Quercetin *	-
**40**	70	28.63	265, 367	285.0385	M-H	C_15_H_10_O_6_	-	Kaempferol *	-
**41**	73	29.25	253, 367	315.0484	M-H	C_16_H_12_O_7_	271.01	Isorhamnetin *	-

^a^ Numbers of identified compounds in the BPI chromatogram (Figure 2). ^b^ Corresponding peak number on the liquid chromatogram (Appendix A). ^c^ Based on previous literature, substituent positions of some compounds have been preliminarily deduced, while substituent positions of other compounds have not been determined. ^d^ Compounds with * were identified with reference standards.

**Table 3 molecules-26-07189-t003:** Antioxidant activity of sea buckthorn berries.

	DPPH (mmol TE·g^−1^)	ABTS (mmol TE·g^−1^)	FRAP (mmol TE·g^−1^)
**W1**			
L	71.80 ± 2.94 ^a^	97.95 ± 1.10 ^a^	67.83 ± 0.14 ^a^
H	17.05 ± 0.20 ^c^	19.80 ± 0.62 ^c^	23.34 ± 1.38 ^bc^
S	12.09 ± 0.74 ^d^	21.95 ± 1.36 ^c^	19.89 ± 1.18 ^c^
D	22.35 ± 1.36 ^b^	26.99 ± 1.24 ^b^	27.07 ± 2.33 ^b^
**W2**			
L	86.30 ± 1.71 ^a^	89.75 ± 2.14 ^a^	61.06 ± 0.70 ^a^
H	17.24 ± 0.63 ^c^	21.70 ± 0.62 ^c^	24.85 ± 1.88 ^b^
S	27.18 ± 1.36 ^b^	26.90 ± 0.24 ^b^	20.24 ± 0.67 ^c^
D	29.27 ± 0.74 ^b^	28.30 ± 1.78 ^b^	25.86 ± 2.04 ^b^
**ZYH**			
L	32.11 ± 1.06 ^a^	34.60 ± 1.12 ^a^	29.14 ± 1.31 ^a^
H	4.85 ± 0.09 ^d^	8.81 ± 0.22 ^c^	7.88 ± 0.15 ^c^
S	6.92 ± 0.12 ^c^	8.96 ± 0.52 ^c^	8.07 ± 0.46 ^c^
D	8.68 ± 0.18 ^b^	11.56 ± 0.60 ^b^	11.71 ± 0.73 ^b^
**SQH**			
L	14.58 ± 0.25 ^a^	18.59 ± 0.61 ^a^	18.52 ± 1.06 ^a^
H	4.36 ± 0.07 ^b^	9.14 ± 0.87 ^b^	8.16 ± 0.12 ^bc^
S	3.68 ± 0.25 ^c^	5.60 ± 0.14 ^c^	6.30 ± 0.88 ^c^
D	3.64 ± 0.22 ^c^	6.22 ± 0.28 ^c^	9.75 ± 0.17 ^b^
**SG**			
L	13.12 ± 0.69 ^a^	17.70 ± 0.92 ^a^	16.54 ± 1.24 ^a^
H	4.62 ± 0.04 ^b^	6.37 ± 0.20 ^b^	7.49 ± 0.44 ^b^
S	3.33 ± 0.19 ^c^	5.98 ± 0.51 ^b^	7.67 ± 0.77 ^b^
D	4.35 ± 0.07 ^b^	5.79 ± 0.36 ^b^	8.36 ± 0.34 ^b^
**C**	4.52 ± 0.17		

Values are expressed as the mean ± standard deviation, n = 3. Significant differences between different drying methods of each variety (*p* < 0.05) are marked with a–d.

**Table 4 molecules-26-07189-t004:** Correlations between antioxidant activity and phenolic compounds.

	DPPH	ABTS	FRAP
DPPH	-	0.982 **	0.970 **
ABTS	0.982 **	-	0.987 **
FRAP	0.970 **	0.987 **	-
TPC	0.980 **	0.986 **	0.988 **
TFC	0.577 **	0.664 **	0.727 **
GA	0.740 **	0.713 **	0.734 **
PA	−0.496 *	−0.490 *	−0.548
RU	0.352	0.391	0.405
Q3G	0.057	0.078	0.064
I3N	0.444 *	0.546 *	0.547 *
K3R	0.397	0.497 *	0.541 *
K3G	0.253	0.352	0.415
I3R	−0.170	−0.180	−0.212
I3G	−0.215	−0.216	−0.247
QE	−0.328	−0.325	−0.376
KA	−0.483 *	−0.457 *	−0.495 *
IS	−0.460 *	−0.452 *	−0.502 *

** Pearson correlation at *p* < 0.01; * Pearson correlation at *p* < 0.05. The larger the correlation coefficient is, the redder the cell color is; the smaller the correlation coefficient, the greener the color.

## Data Availability

Not applicable.

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
