# Peer review of "Impact of Drying Methods on Phenolic Components and Antioxidant Activity of Sea Buckthorn (*Hippophae rhamnoides* L.) Berries from Different Varieties in China"

_molecules, 2021, doi:10.3390/molecules26237189_

Round 1

Reviewer 1 Report

Li et al. describe the effect of 4 drying methods on the component contents and antioxidant activity of 21 sea buck thorn berries from China.

The study is well organized, performed and analyzed. Although it is well known that the lyophilization is the most gentle drying method, the study provides a valuable practical guide for the processing method selection and utilization of the Q-markers for quality control of the herb.

Minor revision:

rows 423 - 428: The legends of Figures 3 and 4 have been swapped.

Author Response

Thank you very much for your kind advice! We are sorry for this mistake. In our uploading WORD file (manuscript.docx), the legends of Figures 3 and 4 were correct. Unfortunately, the order of Figures 3 and 4 was changed when converting WORD to PDF on the website. I have uploaded the correct PDF and WORD files. We apologize again for our mistake.

Reviewer 2 Report

The paper of Chun-Nian He et al. on the impact of drying methods on phenolic components and antioxidant activity of sea buckthorn berries present interesting results that deserve publication.

However, most of the results are expected.

  • Wild-type berries, which are smaller than cultivated berries (line 100), have higher TPC and TFC.
  • The concentration in the final product is also more impacted by the origin (wild vs. cultivated) than by the drying method.
  • Obviously, freeze-drying is better than sun drying
  • Samples from the same sea buckthorn seeds gave similar compositions (ZYH and SQH).
  • Antioxidant activity is related to TPC (section 3.5)

As such the table 2 and Table 3 could be moved into the supplementary information. Furthermore, the value listed in table 3 (m/Z, adducts, MS/MS fragment) are useless. CAS number could be more valuable. The general structure of the flavonol derivatives, quescetin derivatives, kaempferol derivatives, isorhamnetin derivatives, syringetin derivatives, and aglycons would be more interesting for the reader.

Section 3.5 could be also shortened.

However, the use of UPLC-DAD data for principal component analysis is very interesting to discriminate between the drying methods. However, in my pdf, figure 3c and 3d are missing, and new figures are present in figure 4.

In my opinion, the results are worth publication, but the parts dealing with the expected results should be reduced, and the new results (PCA) should be developed and corrected.

Reviewer 3 Report

This manuscript by Li et al. presents an assessment of four varieties of sea buckthorn berry, a plant that has several biological activities, and is used in medicine as well as in the cosmetic and food industries. This berry is a species of great importance in countries such as China, Russia, Mongolia and Turkey. Several bioactive substances and nutrients have already been described in different varieties of the plant, with emphasis on antioxidant compounds. Sea buckthorn berry drying is essential for a variety of applications. The authors evaluated the content of antioxidant metabolites in five varieties of sea buckthorn submitted to different drying methods. They have also identified the main compounds by UPLC-PAD- 87 Q/TOF-MS. I consider the theme of the work quite relevant, due to the academic, commercial and social importance of the results obtained. The methods were properly utilized, and the results were fully discussed. Results were fully analyzed and statistical meaning was discussed. I recommend accepting this manuscript, after minor modifications, as follows.

Introduction.

Introduction gives a good description of the plant, including uses, importance, biological potential, varieties, etc. An overview with pros and cons of the drying processes used during the processing of this species is also presented. The literature cited is pertinent and very useful for visualizing the various aspects addressed by the authors. Some minor corrections:

Line 59. Hippophae rhamnoides ssp. sinensis exhibits

Line 89. components

Methodology.

Line 99. You mention that “The berries of these three cultivars are larger than those from wild sources” If possible, give us a better idea of the proportion. Are they approximately twice the size? Three times? Little larger? In fact, I would appreciate to see a picture of the berries in the manuscript; however, this is only a suggestion.

Line 102. Please, inform whether the commercial sample used as reference was bought in a market or if it is an industrialized product.

Line 123. Unity “C” is missing, please correct to 25 ºC

Line 143. W/V = w/v?

Line 159.The method/mobile phase utilized for UHPLC was very well chosen, resulting in good chromatograms (Supplementary materials)!

Line 167-170. Why the standards were prepared in different concentrations? What “appropriate concentrations” mean? Please, clarify.

Results and discussion

Line 219. Although the authors say that there was a significantly higher difference between their results and the literature (Sytařová et al., 2020), please, check if the unity in line 219 is really in kg and not in g.

Line 245. I was confused here, as you mentioned “mature commercial variety”. Please, inform in the methodology section if the berries from the wild samples studied were also mature.

Table 3 (line 332). What is the meaning of the numbers after some molecular formulas (for example, Quercetin-O-dihexose C27H30O1  7; I did not catch the meaning of the number 7 after the formula. It happens for other compounds). Please, clarify adding an explanation at the bottom of the table.

Line 357. A tentative elucidation based on mass spectra data is interesting and valid. However, I suggest to add the information from lines 356 and 357 at the bottom of Table 3. I spent some time on the table, to understand how the assignment was done and this explanation would have helped me, if it was at the bottom of the table.

The discussion on how different sugar moieties were identified, specific fragments found and considerations regarding retention times are very helpful for readers working in correlated structure elucidation works.

In general, the results are useful because some of the varieties studied by authors have phenolic and flavonoid contents significantly higher than those of other countries (Finland and Slovakia). The results show that the content of phenolic compounds is maintained when drying is done by free drying, while too many processes, which require heating or long periods of processing lead to a decrease in these antioxidants. In general, native species were better sources of antioxidants than the commercial sample evaluated concomitantly in the study. Some compounds were detected and quantified in the samples and several other compounds were tentatively identified by UPLC-PDA-Q/TOF-MS, to provide a wider profile of compounds present in the plants studied. The discussion on marker compounds in berries subjected to different drying methods provided useful information and is an elegant piece of work.  

Conclusion. This section summarizes results obtained and they have an important reference to Chinese Pharmacopoeia (2020). The perspectives of the work are well presented.

Supplementary material. In this file the authors present some chromatograms and tables to be used as reference to some points raised in the main manuscript. It seems alright.

Reviewer 4 Report

After reading this article I come here by the following points.

The article overall is well described, the results are good, the study conducted is interesting but before final acceptance, I would prefer for some minor suggestions and corrections.

  1. Initially, there is a lot of typos and grammatical serious mistakes. Kindly read the paper from start till end line by line to remove the errors. Fix these errors. [Line:118, Four dry methods were used. . .. It should be "Four drying methods were used"]. [LIne 122-124 are very confusing kindly rewrite it with a proper way specially point 4. The berries were kept in a ventilated and dark place, and other conditions were the same as sun drying. . .. "When the berries were placed in a dark place then how were the conditions same as sun Drying? you can write the other conditions were same like point 3, don't write sun drying, it makes it more confusing.
  2. LIne 306; The authors said that "there were still other high peaks that were not identified" Can I know the reason why the other peaks were not identified?
  3. The discussion portion should be separately discussed after the results section, in the results section discussion is done for only some sections but for some sections, discussion is missing. 
